# A GIS-Based Approach to Estimate Electricity Requirements for Small-Scale Groundwater Irrigation

Anna Nilsson [1,*], Dimitrios Mentis [2], Alexandros Korkovelos [3] and Joel Otwani [4]

1 NewClimate Institute, 10119 Berlin, Germany
2 World Resources Institute, Washington, DC 20002, USA; Dimitrios.Mentis@wri.org
3 World Bank, Washington, DC 20433, USA; akorkovelos@worldbank.org
4 GIZ Promotion of Renewable Energy and Energy Efficiency Program Uganda, Kampala 256, Uganda; otwanijoel@gmail.com
* Correspondence: a.nilsson@newclimate.org

**Abstract:** Access to modern energy services is a precondition to improving livelihoods and building resilience against climate change. Still, electricity reaches only about half of the population in Sub-Saharan Africa (SSA), while about 40% live under the poverty line. Heavily reliant on the agriculture sector and increasingly affected by prolonged droughts, small-scale irrigation could be instrumental for development and climate change adaptation in SSA countries. A bottom-up understanding of the demand for irrigation and associated energy services is essential for designing viable energy supply options in an effective manner. Using Uganda as a case study, the study introduces a GIS-based methodology for the estimation of groundwater irrigation requirements through which energy demand is derived. Results are generated for two scenarios: (a) a reference scenario and (b) a drought scenario. The most critical need is observed in the northern and southern regions of the country. The total annual irrigation demand is estimated to be ca. 90 thousand $m^3$, with the highest demand observed in the months of December through February, with an average irrigation demand of 445 mm per month. The highest energy demand is observed in the northern part of the study area in January, reaching 48 kWh/ha. The average energy demand increases by 67% in the drought scenario. The study contributes to current gaps in the existing literature by providing a replicable methodological framework and data aimed at facilitating energy system planning through the consideration of location-specific characteristics at the nexus of energy–water–agriculture.

**Keywords:** energy access; geospatial planning; irrigation; energy–water–agriculture nexus; smallholder farming; Uganda; integrated energy planning; energy systems; GIS

## 1. Introduction

The link between access to energy and development has been widely recognised in numerous studies and contexts globally [1]. Energy consumption is highly interconnected with socio-economic development, education, and health and is vital for poverty eradication and socio-economic development [2,3]. In 2019, lack of energy access was a reality for about 587 million people in Sub-Saharan Africa (SSA) (or 53% of the total population), the majority of which live in rural areas [4,5], negatively impacting local economies [3].

A common factor among many developing economies is their relatively high dependence on the agricultural sector, characterised by a low level of productivity and mechanical input compared to other parts of the world. More than 60% of the SSA population depend primarily on agriculture for their living, accounting for a substantial part of the national GDP [6]. Beyond access to energy as a precondition for improving life quality, it has been observed that energy for productive uses has a potentially higher transformative impact on socio-economic development, being strongly correlated with agriculture [7]. The impact on GDP growth from the agriculture sector is expected to be up to 11 times more effective in reducing poverty than any other sector [8]. As such, food security, income generation,

and poverty eradication have been recognised as development aspects tightly interlinked with the level of agricultural productivity [9].

Moreover, a study from the Food and Agriculture Organization (FAO) proposes that agricultural activity should increase by 60% compared to 2006/07 levels to feed the expected projected world population in 2050 [10]. This implies a significant rise in food crop production and increased pressure on the agro-food system [11]. Feeding the growing world population and simultaneously protecting the environment will require more efficient and sustainable agricultural practices.

Characterised by a high level of subsistence farming, meeting such a rise in demand while facing challenges posed by climate change impacts will require the adoption of measures, which increase the resilience and efficiency of small-scale farming. At present, about 80% of SSA's food is produced by smallholder farmers [8], completely or mainly reliant on family labour to grow staple foods for their own consumption, which is often not sufficient to satisfy the family needs [12].

As the impacts of climate change are increasingly emerging, resulting in prolonged droughts with increased severity, the agro-food system is under a rising pressure, causing crop failure and yield losses [13]. As most of the agricultural activity in SSA is rain-fed and comes from smallholder farmers, rainfall is the climatic factor with the highest economic significance [14]. In severe cases, drought can cause production losses leading to the abandonment of farms, resulting in human migration [15].

A key option for smallholder farmers to build resilience toward climate change impacts is irrigation, which can substantially reduce the risk of crop damage or failure in case of an unexpected drought [16] and decreases the vulnerability to erratic rainfall [9]. Beyond the prevention of risk, irrigation could further improve agricultural productivity. Irrigation has been identified as one of the most effective methods for increased agricultural productivity. Depending on the climate, pumped irrigation can contribute to 50–80% higher yields [9]. The yield could be further maximised through irrigation by extending the rain period to match the crop growing season [17]. Moreover, with access to an irrigation system with low operating costs, farmers are more likely to spend money on other inputs that increase productivity. The resulting increased cropping intensity enables farmers to grow a higher variation of crops in their croplands [18].

Yet, in 2014, only about 4% of the cropland in SSA was irrigated—significantly less compared to 39% in South Asia and 29% in East Asia [10]. As such, the majority of farmers in SSA rely on precipitation for their agricultural activities. A key precondition for the expansion of irrigation is access to electricity for groundwater pumping. The lack of access to electricity limits the productivity of smallholder farmers, affecting their ability to diversify their sources of income [7].

Based on this, irrigation could play a fundamental role in building a more resilient and efficient smallholder agriculture sector in SSA. A well-functioning irrigation system requires a reliable power supply for pumping groundwater. For energy planners and policy makers to efficiently design energy supply solutions in a region, it is essential to use spatially explicit information to understand how demand may vary across space and time.

As such, a more inclusive and integrated energy planning approach is needed to unlock the benefits that improved access to electricity could bring to smallholder farmers. This could enable the identification of populations with a high need for assistance, as well as regions severely impacted by climate change and energy poverty [8]. In this direction, geospatial data and analytical tools could be instrumental in accounting for climatic, environmental, and socio-economic parameters in a systematic and integrated manner [19].

### 1.1. Aims and Objectives

Using Uganda as a case study, we aim to: introduce a bottom up methodology to estimate the spatial distribution of power and energy requirements for groundwater irrigation on a monthly basis across a calendar year. This is achieved through (a) estimating

the spatial distribution of irrigation water requirements from a water balance exercise and (b) estimating the power and energy demand based on irrigation water requirements.

The results are assessed through a scenario analysis to gain an improved understanding of the effects droughts may have on electricity demand.

### 1.2. Previous Studies

In the recent literature, numerous studies highlight the inter-linkage between energy access, agriculture, and development, as well as the importance of irrigation for poverty reduction and climate change resilience. Examples of such are [7,9,10,20]. However, the literature does not provide a methodological framework for generating datasets that could facilitate its implementation. Such datasets should take into account aspects from various disciplines.

Firstly, water irrigation requirements, which vary from one location to another, must be estimated. That involves the quantification of a set of parameters such as the effective rainfall. A standardised methodology for the estimation of the effective rainfall is proposed by [21]. Additionally, the software AquaCrop, developed by the FAO, provides a solid methodology for estimating crop water requirements on a highly detailed level. Both of these methodologies are applied in the study.

Secondly, the geospatial power demand to deliver the irrigation water must be estimated, In [20], a water balance exercise is used as a basis to estimate irrigation electricity demand on a geospatial basis in Tanzania. A similar methodology is applied in [22–24]. Those studies do not, however, consider the potential impacts of different soil types or the implications for irrigation water and electricity demand induced by a drought. Ref. [16] investigates the geographical suitability for solar PV-based water pumping for irrigation in Ethiopia. Similarly, [25] identifies high-priority areas for electrification in Uganda within the energy–agriculture nexus but does not quantitatively estimate the irrigation electricity demand. In [26], the power demand for groundwater pumping is estimated for different times of operation, but the spatial dimension is not considered.

This study aims to contribute to the existing literature by building on existing studies to develop a methodological approach for estimating the spatial electricity demand for groundwater pumping for irrigation and the implications on that demand induced by droughts. The study is conducted through a case study on Uganda of which the results can be replicated for other areas.

## 2. Materials and Methods

This section presents the stepwise methodological approach arriving at the geospatial irrigation water and power demands attributed to the reference crop mix. This is preceded by a description of relevant background information related to the socio-economic status and agricultural sector of the study area.

Figure 1 provides a simplified overview of the methodology and geospatial and non-geospatial input data requirements for the estimation of irrigation water requirements and the subsequent derivation of energy and power demand. The geospatial analysis is performed in QGIS, which is an open-source Geographic Information System software, which enables the combination and analysis of geospatial datasets [27].

### 2.1. The Case Study of Uganda

2.1.1. Socio-Economic Status and the Relevance of the Agriculture Sector

Uganda is a landlocked country located in East Africa. As of 2019, it had a population of 44 million 76% of which residing in rural areas [28]. The country has one of the youngest and most rapidly growing populations globally with an average age of 16 years and an annual population growth rate of 3.6%—significantly above the SSA average [29]. In 2016, about 41% of the population lived below USD 1.90 per day, with the highest concentration of poor people settled in rural areas, relying on agricultural activities for their livelihoods [30]. The electrification rate in Uganda is low; in 2019, 41.3% of the population had access, while the corresponding rate in rural settlements was 31.8% [5].

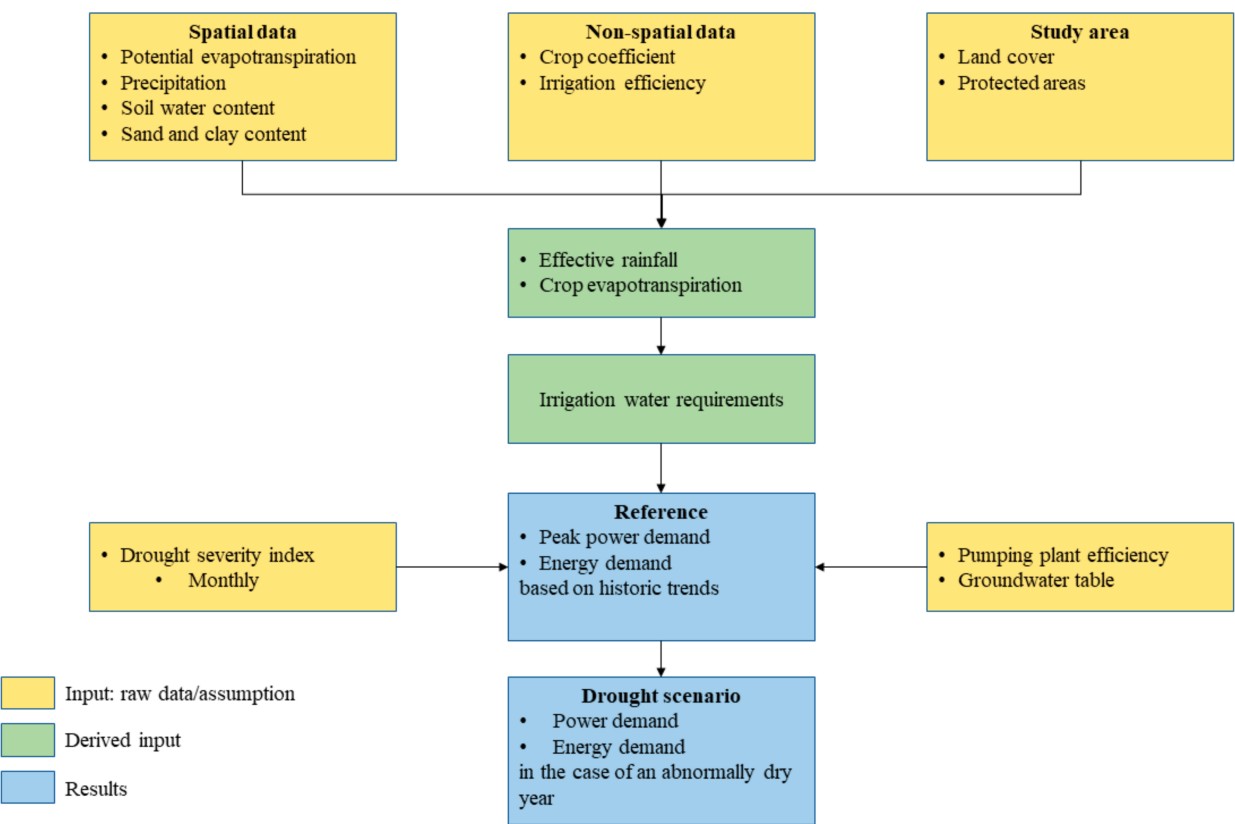

**Figure 1.** Methodological flow chart.

The agriculture sector is the main employer, accounting for 70% of the total workforce [31]. It is dominated by smallholder farmers on average owning 1–2 hectares of land [31]. Agriculture output contributes to about 25% of GDP, 50% of exports [32], and has been identified as the most impactful sector for poverty eradication [29]. Specific importance is recognised within the production of cash crops—coffee, in particular—contributing to more than 20% of the country's export revenues [33]. Agricultural income growth in Uganda is found to be more strongly linked to consumption growth compared to other sources of income growth and contributed to a 79% poverty reduction between 2006 and 2013 [29].

Access to energy is identified as a critical issue for Ugandan agriculture [30]. The sector has a low mechanization rate and is heavily dependent on rainfall and fertile soils [31]. Rainfalls are of bimodal character (i.e., two well defined rainy seasons within a year) in 70% of the country, gradually becoming uniform with increasing distance from the equator [14]. Rainy seasons are historically observed from March through May and September through November, while typically drier seasons are from December through February and June through August [14]. Rainfall has been identified as the most sensitive climate variable and main determinant affecting social and economic activities. Previous studies have shown that 7 out of 16 regions in the country are prone to drought [14]. Since 1960, increased average temperatures and unpredictable and poorly distributed rainfall have been observed, as well as more frequent occurrence of extreme weather events such as droughts [31]. These climatic changes adversely affect agricultural productivity levels, falling well below the potential [14].

### 2.1.2. Farming Systems and the Relevance of Cash Crops

Two major farming systems are practised in the country: the mixed maize system in the northern half and highland perennial in the southern half [34]. Highland perennial farming systems in Uganda hold the highest rural population density, with an extensive

prevalence of poverty. Households have an average cultivated area of just below one hectare [34]. The mixed maize farming system is the most important in Eastern Africa. The farm density is moderately high with an average farm size that usually is less than 2 hectares. The farming system may be used in both bimodal as well as unimodal (one annual rainy season) areas. The output crops are typically grouped after cash crops and staple crops. Coffee is grown as a cash crop in both highland perennial and maize mixed farming systems but is typically intercropped with various crops depending on what farming system is practised. It is rarely seen that a farmer solely produces coffee without intercropping [34].

In [34], the mixed maize and the irrigated systems are highlighted for their potential poverty reduction and agricultural production growth, yet the mixed maize system is highly vulnerable to droughts. A top priority for that farming system is diversification by shifting from low-value crops, such as maize, to high-value crops, such as coffee. This is ultimately dependent on reliable and just access to resources, such as land, water, and energy [34].

As of 2016, about 1% of the cropland in Uganda was irrigated [35]. In 2010, drought resulted in 38% and 36% losses in production for beans and maize, respectively. Such occurrences put many people on the margin to meet their minimum food needs [31].

Given the significance of coffee as a cash crop in the Ugandan agriculture and economic sector, and its potential to further improve livelihoods, coffee is used as the reference crop in this study. Throughout the study, the reference crop is assumed to be intercropped with banana and groundnut—two common staples in the study region—and serves as the basis for the estimation of irrigation water requirements. The study exclusively considers groundwater pumping.

### 2.2. Definition of the Study Area

The study is constrained by the administrative geographical boundaries of the Republic of Uganda. Within these boundaries, the study area is further defined by land cover use and protected areas, as described in the next section.

### Identification of Crop Land

Land cover maps from the land cover classification CCI-LC project developed by the European Space Agency (ESA) are based on the UN Land Cover Classification System (UN-LCCS) [36]. The study considers irrigation for cropland in Uganda, while the following categories are counted as the study area:

- Rain-fed cropland;
- Mosaic cropland (>50%)/natural vegetation (<50%);
- Mosaic natural vegetation (>50%)/cropland (<50%).

These areas are extracted from the original raster layer with the QGIS raster calculator. The complete share of the rain-fed cropland is considered, being land devoted to agriculture. Of the mosaic cropland, 60% is assumed to be cropland, while the mosaic natural vegetation is assumed to be covered up to 40% by cropland. The aforementioned are educated assumptions made by the authors. The study area is further constrained by excluding protected areas [37] (Figure 2). The study draws on openly accessible datasets, presented in Table 1.

A reference scenario represents current conditions and is characterised by a groundwater depth of 7 m and a daily time of operation of 8 h. Considering the low resolution and quality of used groundwater depth data, the whole study area falls under one depth category of 0–7 m. Therefore, in the reference scenario, the depth is assumed to be 7 m across the study area. The impact of these parameters is later analysed in a sensitivity analysis (see Supplementary Materials). The first step of the process is the estimation of irrigation water requirements. Based on the results, the pumping power and energy requirements are calculated.

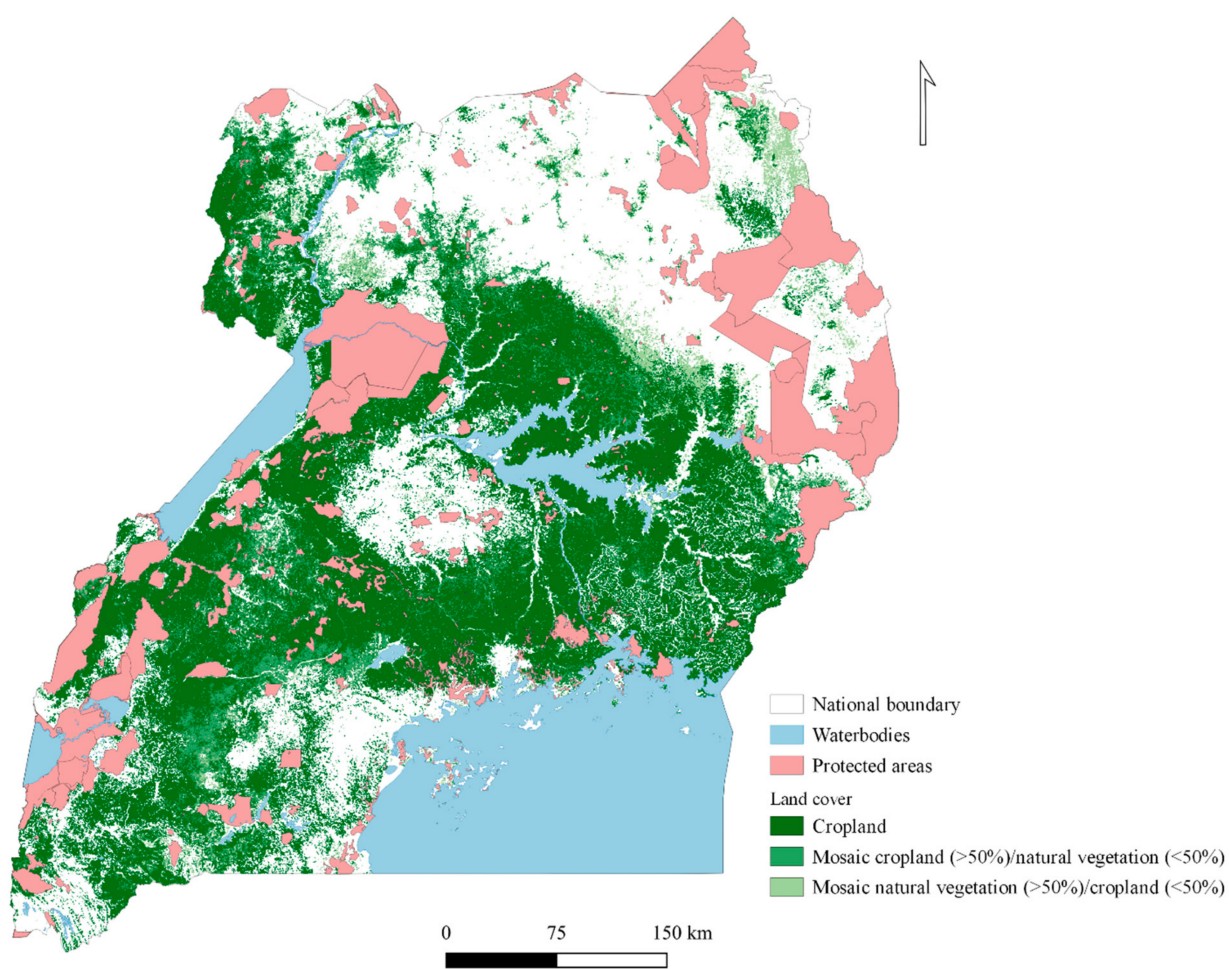

**Figure 2.** Definition of the study area. The study area is restricted by protected areas and limited to crop land.

**Table 1.** Spatial data parameters and sources.

| Methodological Step | Dataset | Source | Spatial Resolution |
|---|---|---|---|
| Study area | Land cover (ESA) | European space agency (ESA) [36] | 300 m |
| | Protected areas | World database on protected areas (WDPA) [37] | polygon |
| | District boundaries | Energy Sector GIS Working Group Uganda [38] | vector |
| Irrigation water requirements | Potential evapotranspiration | CGIAR-CSI [39] | 30 s |
| | Precipitation | WorldClim 2.0 Beta version 1 [40] | 30 s |
| | Soil water holding capacity | ISRIC Africa soil information service (AfSIS) [41] | 1 km |
| | Soil clay content | ISRIC Africa soil information service (AfSIS) [41] | 250 m |
| | Soil sand content | ISRIC Africa soil information service (AfSIS) [41] | 250 m |
| Power and energy demand | Groundwater table | British Geological Survey [42] | 5 km |
| | NDVI percentile | Princeton Climate Analytics [43] | 5 km |
| | Sub-county boundaries | GIS energy sector working group [44] | polygon |

### 2.3. Estimating Crop Water Requirements

In this section, existing methods for estimating irrigation water requirements are presented, followed by the methodology introduced in this study and the characteristics of its components. A graphic overview of the GIS input data is presented in the Supplementary Materials.

### 2.3.1. Existing Methodologies for Estimating Irrigation Water Need

In the existing literature, there are several different methodologies for estimating the amount of required irrigation water, applying various levels of detail. In general, the required irrigation water volume is determined by the level of evapotranspiration (ET), referring to the moisture transferred from the soil and plant to the atmosphere. The evapotranspiration subtracted by the amount of rainfall constitutes the amount of water needed for the crop to stay healthy and proliferate [45]. However, various factors affect the rate of evapotranspiration and subsequently the share of rainfall that is available to the crop. Examples of such aspects are the crop and soil type. Taking these aspects into account, a methodology is developed based on the basic water balance as the core:

$$\text{Irrigation Water Requirements} = \text{Evapotranspiration} - \text{Precipitation} \qquad (1)$$

In [46], the irrigation water requirements (IR) are estimated by subtracting the amount of rainfall by the actual evapotranspiration, which is the evapotranspiration multiplied by a crop coefficient ($K_C$). In other studies, such as [47], the share of effective precipitation ($P_{Eff}$) is also accounted for, which itself can be estimated from a number of different methodologies. These are explained in more detail in Section 2.3.3. The original water balance equation is further developed into:

$$\text{IR} = K_C \cdot \text{ET} - P_{Eff} \qquad (2)$$

where

IR = irrigation water requirements;
$K_C$ = crop coefficient;
ET = evapotranspiration;
$P_{Eff}$ = effective precipitation.

The parameters affecting the variables in this equation vary depending on climatic and ecological conditions. In the following section, the variables affecting IR and their behaviour are described in detail.

### 2.3.2. Evapotranspiration and the Crop Coefficient

Evapotranspiration refers to the mass transfer of water exchanged by the soil and crop with the atmosphere. Depending on temperature, sunlight, relative humidity, and wind speed, water is transferred from the soil, via evaporation, and from the crop via transpiration.

There are several different methods used for the estimation of a reference evapotranspiration [48]. A standard method used by the FAO for estimating potential evapotranspiration (PET) is the Penman–Monteith equation. PET is an estimation of the ability of the atmosphere to transfer water from soil and crops, considering a reference crop (grass) growing under optimal conditions [49].

Thus, the site-specific climatic influences on ET are considered in PET. However, also, crop-specific factors need to be accounted for, most commonly through the introduction of the $K_C$. The actual ET of a crop or mixture of crops, $ET_C$, is, thus, given by the relation given in Equation (3):

$$ET_C = K_C \cdot \text{PET} \qquad (3)$$

The $K_C$ may vary for a specific crop depending on which stage of its growth cycle it is in and the availability of water in the soil and atmosphere. The $K_C$ tends to increase with the availability of water, resulting in a high $K_C$ under periods of ample rain or irrigation and in a low $K_C$ in the case of drought [50].

Coffee is grown across the country but is mainly concentrated to the south and southwest, central, and eastern as well as in the northwest of Uganda [51]. For simplicity, the same intercrops are considered within the study area. The choice of the type of intercrops quantitatively mainly affects the results through the $K_C$ and can be easily adjusted in other studies.

For this study, the respective $K_C$ values for coffee, banana, and groundnuts are collected from the FAO tabulated data for three different growth stages [52]. The $K_C$ for coffee typically ranges between 1.05 and 1.1 for a crop with weeds and between 0.9 and 0.95 for clean weeded crops. However, there is a disagreement with other studies, where the KC for coffee is estimated to span from 0.5 in the dry season to 0.8 during the rainy season. In this study, an average value of the results from the different sources is used. Since irrigated crops are considered, the $K_C$ may only reach its dry period values in cases when irrigation and precipitation is absent and reaches its peak value when the precipitation ($P_T$) is higher than PET [53]. Those are the conditions that define which $K_C$ is applied in the model.

Food staples are often grown under canopies of coffee and/or banana trees with a random spatial distribution [54]. It is recommended to intercrop coffee with bananas and legumes such as groundnut—commonly one banana tree per four coffee trees [55]. It is, therefore, assumed that 75% of the field is covered by coffee crops and that the remaining 25% of the field is a mixture of banana trees and groundnut crops, leading to the relation in Equation (4):

$$K_{C,tot} = 0.75 \cdot K_{C,C} + 0.25 \cdot K_{C,BG} \tag{4}$$

where

$K_{C,tot}$ refers to the combined crop coefficient of coffee and intercrops.
$K_{C,C}$ refers to the crop coefficient of the coffee crop.
$K_{C,BG}$ refers to the combined crop coefficient of the banana and groundnut crops.

Table 2 lists the individual, average, and final $K_C$ values for the crops considered. $K_{C,\,Tot}$ depends on the expected wetness or dryness of the month and is determined by the level of PET and $P_T$. Under wet circumstances, when $P_T$ is higher than PET, the $K_{C,Peak}$ is applied; otherwise, the $K_{C,Reference}$ is used.

**Table 2.** Crop coefficients with data for coffee, banana and groundnut from [52].

| Crop | $K_C$ | |
|---|---|---|
| | **Reference** | **Peak ($P_T$ > PET)** |
| Coffee | 0.90 | 1.20 |
| Banana | 0.85 | 1.20 |
| Groundnut | 0.88 | 1.15 |
| | **Derived $K_C$** | |
| Average, banana, and groundnut ($K_{C,avg\,BG}$) | 0.86 | 1.18 |
| Overall average ($K_{C,tot}$) | 0.89 | 1.19 |

Monthly average spatial data from 1950 to 2000 of PET are collected from the Global Aridity and PET Database, based on input climate data from WorldClim [56]. The $ET_C$ is derived from PET data and the $K_C$ values.

### 2.3.3. Effective Precipitation

Not all received precipitation is available to the crop for consumption. Some of the precipitation will evaporate to the air even before striking the soil—increasing the humidity of the air, resulting in a declining $ET_C$. The water that reaches the soil will subsequently either infiltrate into the soil, stagnate on the surface, or flow over the surface as runoff. Some of the water that infiltrates the soil will percolate below the root zone; the stagnating water will either evaporate or, after some time, infiltrate into the soil. Such losses are

accounted for in the effective precipitation ($P_{Eff}$). In short terms, $P_{Eff}$ can be referred to as the amount of water available to the crop after subtracting mentioned losses [45].

The fraction of effective precipitation depends on several factors, which makes its estimation somewhat complex. One important factor is the initial conditions of the soil, the soil type, and its water holding capacity. In general, the finer the soil texture, the greater is the water holding capacity and, thus, also, the fraction of effective precipitation.

There is no standard method recognized in the literature for estimating $P_{Eff}$. In [21], various existing methods are assessed, and, based on their identified limitations, a new approach is proposed. This approach is dependent on crop rooting depth, climatic conditions, and whether the studied crop is a dry- or wet-land crop. And it can be used as a generalized form in simulation models. Following this, the effective precipitation in this study is determined by multiplying the total amount of precipitation with a factor, C, which depends on the climate, soil, and whether the considered crop is deep- or shallow-rooted. The coffee crop is a shallow-rooted, dry-land crop [57]. For shallow-rooted crops, C ranges from 0.8 to 0.95.

Subsequently, soil water holding capacity (WHC) data for the region are collected from [58], ranging from 6 to 12% within the study area. The greater the WHC, the greater the fraction of effective rainfall will be [59]. Hence, C and the WH are divided into three classes and linked, as demonstrated in Table 3. In such circumstances when $P_T$ is greater than PET, it is assumed that $P_{Eff}$ is equal to the total amount of crop water consumption, and no irrigation is required.

**Table 3.** Determination of the coefficient of efficient precipitation.

| WHC (%) | $6 \leq$ WHC $< 8$ | $8 \leq$ WHC $< 10$ | $10 \leq$ WHC $\leq 12$ | - |
|---|---|---|---|---|
| C | 0.90 | 0.85 | 0.80 | - |
| $P_T$ | $\leq E_{TC}$ | $\leq E_{TC}$ | $\leq E_{TC}$ | $> E_{TC}$ |
| $P_{Eff}$ | $C \cdot P_T$ | $C \cdot P_T$ | $C \cdot P_T$ | $E_{TC}$ |

### 2.3.4. Irrigation Efficiency

As with the precipitation, not all water supplied by irrigation is useful to the crop. Irrigation water losses can be divided into two groups resulting in two efficiencies: conveyance and field application. The conveyance efficiency ($\varepsilon_C$) primarily depends on the length of the canals and the soil type. The sandier the soil is, the more water is lost. Further, larger irrigation systems tend to lose more water. The conveyance efficiency is calculated from indicative tabulated values [45]. Since this study considers small-scale irrigation, indicative values for medium to short canal lengths are used. The conveyance efficiency is calculated for the three different soil types of clay, loam, and sand (Table 4). The definitions for different soil types are explained in the next section.

**Table 4.** Conveyance efficiency depending on the soil type.

| Soil Type | Sand | Loam | Clay |
|---|---|---|---|
| Medium canal | 0.70 | 0.75 | 0.95 |
| Short canal | 0.80 | 0.85 | 0.95 |
| Average | 0.75 | 0.80 | 0.95 |

The field application efficiency ($\varepsilon_A$) mainly depends on the irrigation method used. In this study, an indicative average value of 0.75 is applied [45]. The overall irrigation efficiency for each soil type is calculated from Equation (5) and is presented in Table 5.

$$\varepsilon_{Irr} = \varepsilon_C \cdot \varepsilon_A \tag{5}$$

**Table 5.** Irrigation efficiencies [28,50].

| Efficiency | Sand | Loam | Clay |
|:---:|:---:|:---:|:---:|
| $\varepsilon_{Irr}$ | 0.56 | 0.60 | 0.71 |

### 2.3.5. Definition of Soil Type

Spatial data on clay and sand content are collected from the African soil information system (AfSIS) [60,61]. Depending on the share of sand and clay, the soil is categorised as clay, loam, or sand. The soil types are defined using the SPAW—a water budgeting tool developed by USDA, which enables the simulation of soil water characteristics such as soil type [53]. The respective soil type definitions are presented in Table 6, where θ represents the volume (%) of the top 30 cm of the soil. Each soil type is defined in GIS as raster layers.

**Table 6.** Definition of soil type.

| Soil Type | Definition of Soil Type |
|:---:|:---:|
| Clay | $\theta_{Clay} \geq 35\%$ and $\theta_{Sand} < 45\%$ |
| Loam | $10\% \leq \theta_{Clay} < 35\%$ and $\theta_{Sand} < 50\%$ |
| Sand | $\theta_{Sand} \geq 50\%$ and $\theta_{Clay} < 35\%$ |

### 2.3.6. Irrigation Water Requirements

Ultimately, all variables in the irrigation water requirements equation are calculated, and IR is determined from Equation (6):

$$IR = (PET \cdot K_C - P_{Eff})/\varepsilon_{Irr} \tag{6}$$

### 2.4. Estimating Peak Power and Energy Demand

This section presents the methodological approach for calculating the peak power and overall energy demand for irrigation groundwater pumping, based on the IR results from the previous section. An overview of the required input data is followed by energy demand calculations theory and its application in QGIS.

### 2.4.1. Spatial Input Data

Spatial data required for the estimation of the power and overall energy demand include the IR, which is obtained from the derived output from Section 2.2, and the groundwater table, which is a key aspect affecting the required work for pumping. High-quality openly accessible data on groundwater levels in the study region are limited and were only found with a spatial resolution of 5 km, provided by the British Geological Survey (BGS) [42]. The dataset is category based, dividing the depth into the following categories: 0–7; 7–25; 25–50; 50–100; 100–250 m. According to these data, the groundwater table within the study area lies within the first category of 0–7 m. However, from other BGS studies in the region, it is stated that the depth can range from 1 to 45 m, but is typically between 5 and 20 m below the ground surface [60]. Due to this uncertainty, the pumping power demand ($P_{Pump}$) in this study is calculated at the four different depths of 7, 10, 15, and 20 m. The results are presented as a sensitivity analysis (see Supplementary Materials). A similar approach has been used in other studies [16].

### 2.4.2. Non-Spatial Input Data

Non-spatial data do not vary with geographical location and include the overall pumping plant efficiency, the daily time of operation, and the irrigation area. The overall pumping plant efficiency is determined by four technical parameters including the fuel, power unit, transmission, and pump efficiency, which are presented in Table 7.

**Table 7.** Parameters considered in the overall pump efficiency. Data points are sourced from [61].

| Parameter | Range | Value Applied in This Study | Motivation |
|---|---|---|---|
| Fuel efficiency ($\varepsilon_{\text{Fuel}}$) | Typically ranges from 90 to 100% | 100% | No petrol or diesel; no leakages |
| Power unit efficiency ($\varepsilon_{\text{Power}}$) | Ranges from 10 to 35% for petrol and diesel in small pumps, and from 75 to 85% in electric pumps | 75% | The study focuses on electrification |
| Transmission efficiency ($\varepsilon_{\text{Transm}}$) [1] | Generally close to 100% | 100% | |
| Pump efficiency ($\varepsilon_{\text{Pump}}$) | With optimal speed and head ranges between 40 and 80%. Usually near optimal conditions. | 60% | Average value is applied |

[1] The transmission efficiency refers to the transmission from the engine to the pump unit, which, if directly coupled, is close to 100%.

Finally, the overall pump efficiency ($\varepsilon_{\text{Tot}}$) is determined from Equation (7):

$$\varepsilon_{\text{Tot}} = \varepsilon_{\text{Fuel}} \cdot \varepsilon_{\text{Power}} \cdot \varepsilon_{\text{Pump}} \cdot \varepsilon_{\text{Transm}}$$
$$\rightarrow \varepsilon_{\text{Tot}} = 0.45 \tag{7}$$

In [26], four different daily times of operation ($t_{\text{Op}}$) are applied when estimating pumping power demand: 8, 12, 16, and 20 h. The FAO suggests a $t_{\text{Op}}$ of 6 h daily [61]. In this study, it is assumed that the irrigation scheme operates 8 h daily but is further analysed in a sensitivity analysis (see Supplementary Materials). With respect to irrigation area, power and energy demands are calculated per hectare, as the average size of a smallholder farm is assumed 1 hectare. In addition to monthly results, the annual aggregated energy demand per sub-county is also calculated.

### 2.4.3. Basic Physics and Pumping

In order to determine the energy and peak power demand, the work required to transport ground water to the surface point is estimated. The quantity of work is determined by (a) the head, H, which refers to the vertical distance the water is to be lifted; (b) the discharge rate, Q, which refers to volume of water discharged per unit of time; (c) gravity, g; (d) water density, ρ, and ε the efficiency of the pump, $\varepsilon_{\text{Tot}}$. In a first step, Q is calculated, assuming a $t_{\text{Op}}$ of 8 h (Equation (8)):

$$Q\left[\frac{m^3}{s}\right] = \frac{IWR[m^3]}{30\left[\frac{days}{month}\right] \cdot 8\left[\frac{h}{day}\right] \cdot 3600\left[\frac{s}{h}\right]} \tag{8}$$

Based on $Q$, the peak power demand ($P$) is calculated from Equation (9):

$$P[kW] = \left(\frac{Q\left[\frac{m^3}{s}\right] \cdot 9.81\left[\frac{m^2}{s}\right] \cdot H[m]}{\eta_{Pump}}\right) \cdot 10^{-3} \tag{9}$$

Finally, the aggregated monthly power demand ($E$) is determined from Equation (10):

$$E[kWh] = P[kW] \cdot 8\left[\frac{h}{day}\right] \cdot 30\left[\frac{days}{month}\right] \tag{10}$$

The results are generated as raster layers for each month of the year. The raster layers have a granularity of one hectare and are generated for the power and energy demand ($P$ and $E$), respectively. Further, the aggregated annual energy demand of the complete study area is generated.

*2.5. Scenario Analysis*

In the scenario analysis, a drought year is simulated by assuming a lower-than-average precipitation. It aims to assess how the energy demand could be affected in the case of a drought. The level of drought severity is based on a drought severity index derived from historic climatic data.

2.5.1. Definition of a Drought

A drought refers to a period of limited rainfall below the normal level. Normal in this study refers to the monthly average level of precipitation between 1970 and 2000. However, drought can be defined differently depending on discipline. To a farmer, a drought is characterised by the lack of moisture in the crop root zone, unable to meet evapotranspiration demands [62]. The historically relatively well-defined rain periods in Uganda have been interrupted and varied significantly in the last 20 years. While the total amount of rainfall has not changed much, its timing has become increasingly unpredictable, often characterised by prolonged droughts [33].

2.5.2. Methodological Approach

According to [17], crop water stress reaches high severity once the available water meets less than half the crop water demand. In the same study, a dry spell occurs when the effective rainfall is less than 50% of the reference evapotranspiration. The literature provides several drought indices measuring the severity of droughts, differing in suitability depending on the aim of the study. The African Flood and Drought Monitor developed a drought index, which specifically targets agricultural drought and is based on the soil moisture content and the Normalized Differential Vegetation Index (NDVI). It is typically referred to as the NDVI percentile index and measures the severity of agricultural drought on a monthly basis. The NDVI percentile index is calculated from historic climatic data, hydrological modelling, and satellite remote sensing [63]. It is chosen as a suitable indicator to define drought in the study, as it specifically considers agricultural droughts. Other indices, such as the Palmer Drought Severity Index (PDSI), are considered less appropriate, as they are more general and focus on annual changes. Using the results from the reference scenario and spatial NDVI percentile data as inputs, a methodology for a drought scenario analysis in this study is developed.

Monthly spatial data (raster, 5 km) from of the NDVI percentile index (from here onwards referred to as the Agricultural Drought Severity Index, ADSI) are collected for the study area through the period 2003–2008, being the most recent data. The ADSI ranges from 1 to 100, where low values indicate drought conditions. The annual data for each month and each year are added and averaged to identify the areas historically most exposed to agricultural droughts (Figure 3).

Depending on the severity of ADSI, the $P_{Eff}$ will decrease by different factors, which are monitored through a varying coefficient, C, described in Section 2.3.3. The higher the probability/severity of ADSI, the lower C will be (Figure 4). ADSI values from 0 to 60 are considered risk zones, where C varies between 0.2 and 0.4 depending on the severity. A level of ADSI above 60 yields a C value of 0.6, which is still below the reference, where C varies between 0.8 and 0.9.

The effective precipitation in the drought scenario, $P_{Eff,D}$, is calculated from Equation (11):

$$P_{Eff,D} = C \cdot P_T \qquad (11)$$

The irrigation water requirements in the drought scenario ($IR_D$) and the respective peak power and aggregated energy demand ($P_D$ and $E_D$) are calculated according to Equations (6), (9) and (10).

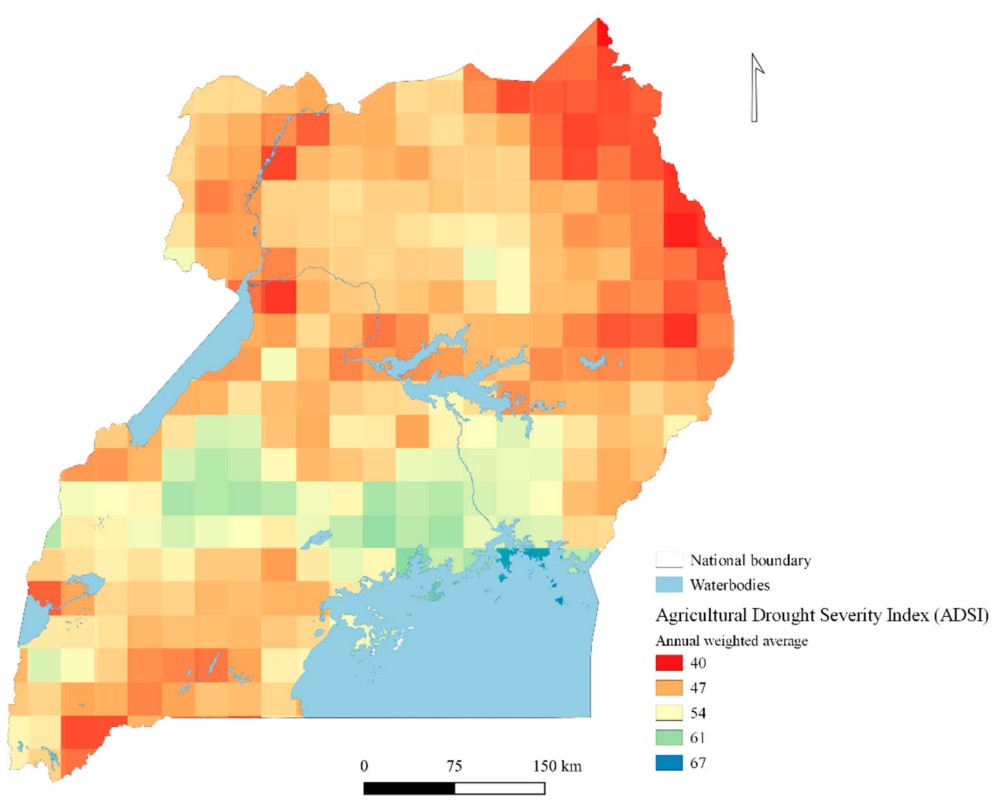

**Figure 3.** The weighted average of ADSI between 2003 and 2008.

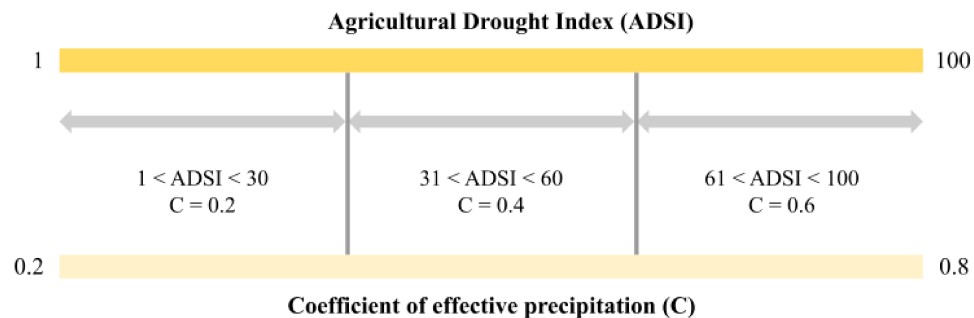

**Figure 4.** C reclassified depending on ADSI; C increases with an increased ADSI.

## 3. Results

This section presents the results for irrigation water requirements, peak power, and energy demand in the reference scenario. Following, the scenario analysis results assessing the impacts of an abnormally dry year are presented, discussed, and analysed. Monthly results are presented in the supplementary materials.

### 3.1. Reference Scenario

3.1.1. Water Requirements for Groundwater Irrigation

The total annual irrigation requirements for the selected crops (coffee, banana, and groundnut) in Uganda are estimated at 90.4 thousand m$^3$. On a national level, groundwater irrigation needs are at their highest between December and February, averaging at 445 mm (Figure 5). Peak irrigation need is observed in January—the driest month of the year in Uganda. On a sub-national level, irrigation needs are particularly high in the northern region of the country (Karamoja and West Nile regions). On the contrary, in the southern parts of the country, high irrigation needs are observed between June and August (Figure 5). Over this period, the average irrigation need across the country is 195 mm.

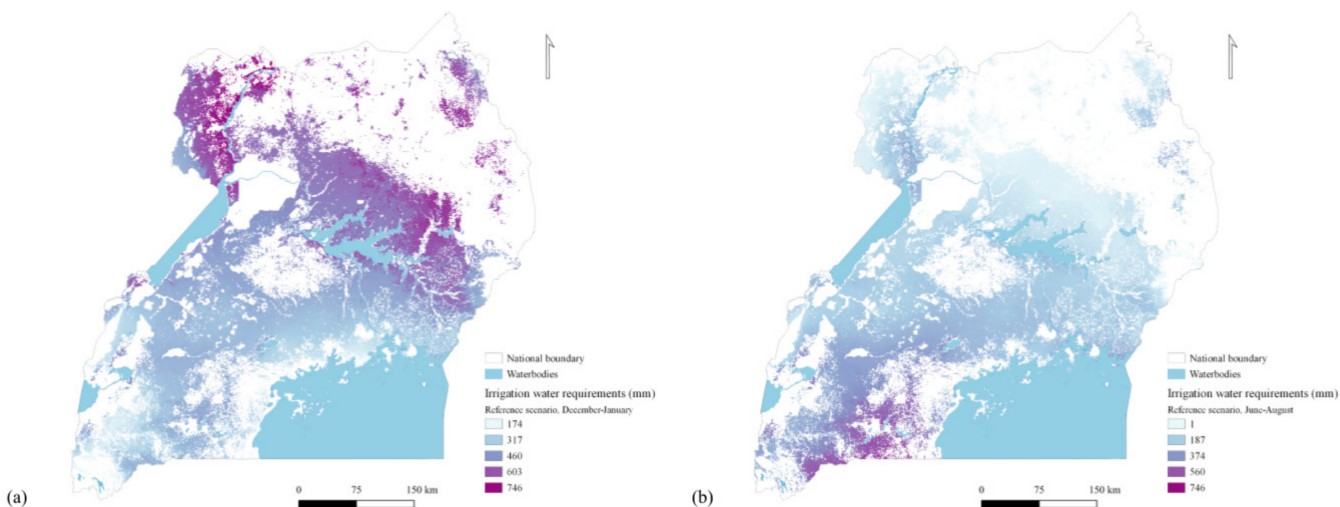

**Figure 5.** Aggregated irrigation water requirements in the reference scenario for the periods of December through February and June through August, respectively. The results are presented as follows: (**a**) December through January and (**b**) June through August. A notably higher demand is observed in the northern part of the study area from December through January, while there is a generally higher demand in the south from June through August.

The overall lowest demand is seen in April through May, when there is no need for irrigation in large parts of the study area, except from in the south, northwest, and northeast regions. September through November, there is a moderate demand across the study area; however, this is slightly higher in the northern and southern regions. Finally, studying the aggregate irrigation water demand over the full year, the highest demand is observed in the northern and southern regions, averaging at 921 mm across the country.

Despite the fact that some regions are in no need of irrigation during some parts of the year, the overall demand never reaches zero across the study country.

### 3.1.2. Peak Power Demand

The peak power demand is presented in units of kW per hectare of each cell. The peak power demand distribution for January, April, July, and October is presented in Figure 6. The power demand varies regionally throughout the year with overall highest intensity in January, which is when the highest demand is observed in the northern region of the country. However, demand intensities of the same levels are observed in other regions at other times of the year. An example of this is the intensity seen in the southern region in the month of July.

Based on this, it can be concluded that irrigation power demand is likely to have a higher impact on the overall power system during the months of December through March, following historic climatic patterns (Figure 7). Nevertheless, local high demand can be expected in various parts of the country throughout the year. At its most, power demand intensity reaches close to 0.5 kW/ha in January in the north. The lowest overall demand is observed in April and May, when large parts of the study area, especially in the central parts, are in no need of pumping power at all. In April, the highest observed power demand is 0.2 kW per hectare.

### 3.1.3. Energy Demand

The behaviour of the energy demand throughout the year follows a similar pattern as the power demand. At its peak, it reaches 118 kWh per hectare in the northern and northwest parts in January. In the wettest months, April and May, the peak energy demand reaches 48 and 45 kWh per hectare, respectively.

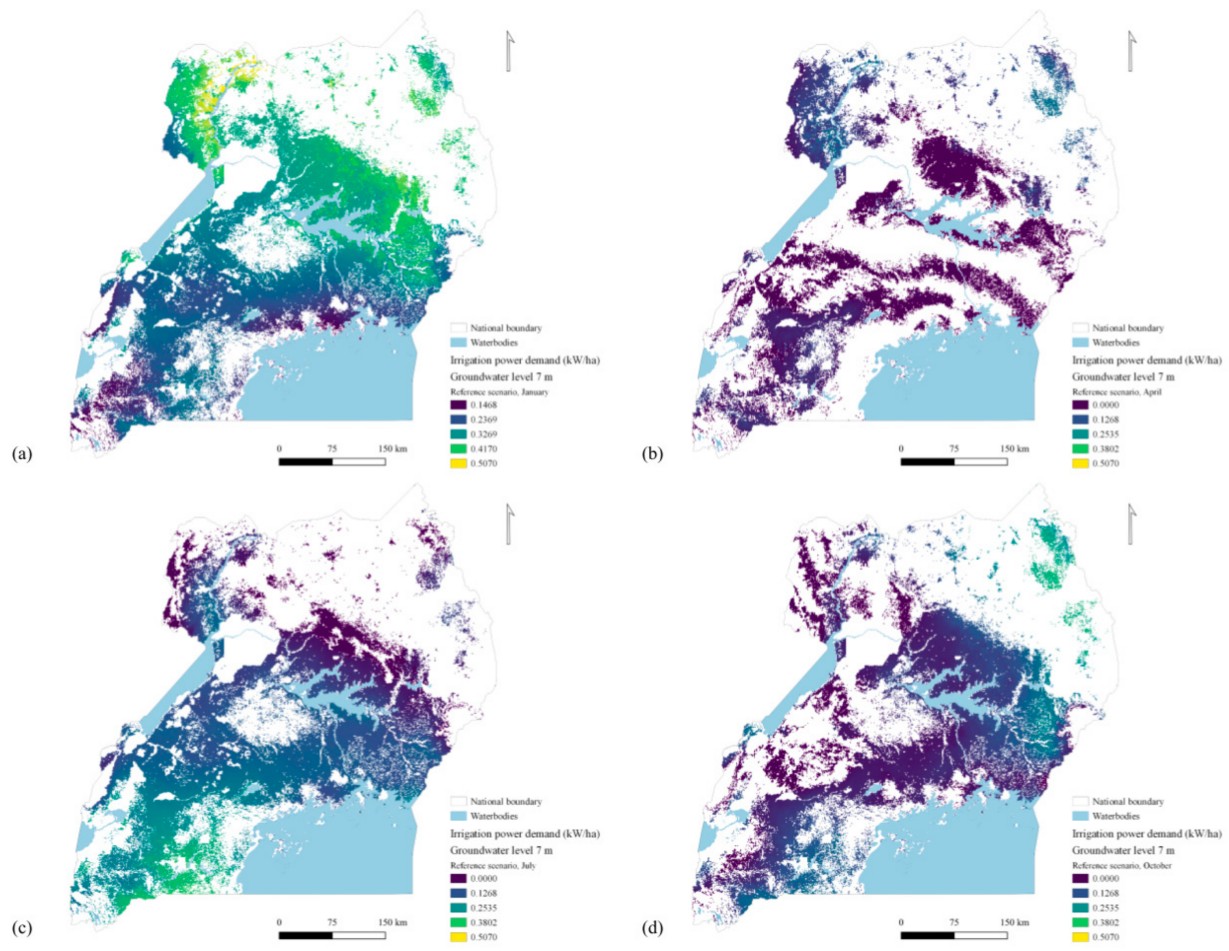

**Figure 6.** Peak power demand for irrigation water pumping in the reference scenario in January, April, July, and October. The results are presented as follows: (**a**) January, (**b**) April, (**c**) July, and (**d**) October. The results indicate a varying demand across the year in terms of scope and intensity.

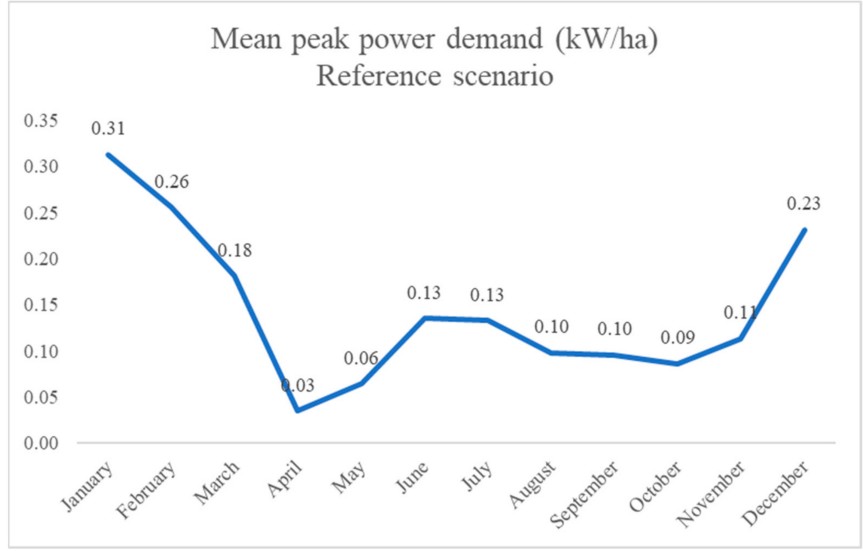

**Figure 7.** Mean peak power demand distribution (kW/ha) for irrigation water pumping throughout the year and study area in the reference scenario. The peak power demand remains rather stable between June and November and peaks in January. The lowest peak power demand is observed in April.

The maximum annual energy demand is also observed in the north, where it reaches 767 kWh per hectare while the generally highest annual energy demand is observed in the northwest, northeast, central-east, and in the south/southeast (Figure 8). The lowest demand is found in the central region of the country, reaching 182 kWh per hectare at its lowest point.

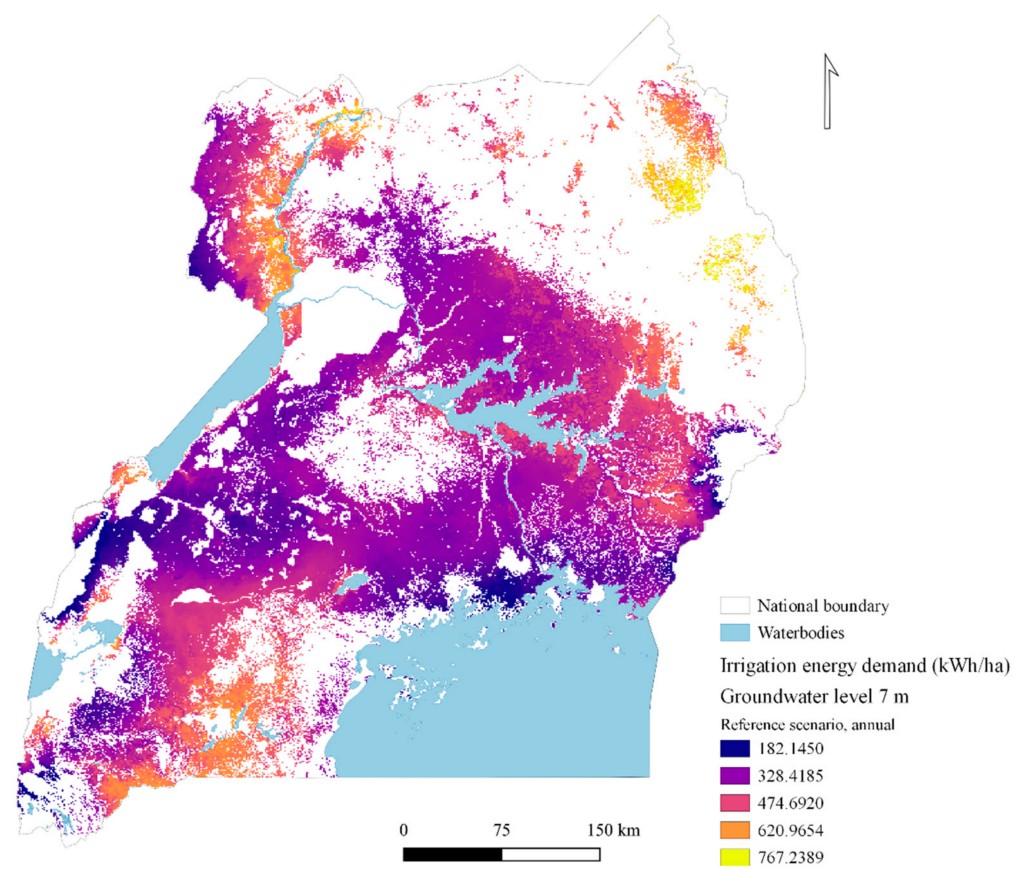

**Figure 8.** Annual energy demand distribution (MWh/ha) for irrigation water pumping, reference scenario.

From studying the annual aggregate energy demand per sub-county, a similar pattern is observed as in the demand distribution per hectare (Figure 9). The highest level of demand reaches 385 MWh in the northeast, Karamoja region. Elsewhere, energy demand is high in the northwest and southeast regions ranging between 14 and 29 GWh per year and sub-county. Logically, the largest levels are observed in the sub-counties with the largest area but are further linked to the energy intensity per hectare.

### *3.2. Drought Scenario*

### 3.2.1. Water Requirements for Irrigation, Drought Scenario

In the drought scenario, the results show a significant rise in overall irrigation water requirements. The average demand increases to 1416 mm, corresponding to a 54% increase compared to the reference scenario. As observed in Figure 10, the simulated drought does not affect the study area uniformly. Areas in the northern and southern part of the study area are more severely impacted and experience a higher relative rise in irrigation water demand. The results, thus, demonstrate that the planning of energy supply could consider adaptive measures by preparing for drought scenarios in regions that are more likely to be severely affected by droughts.

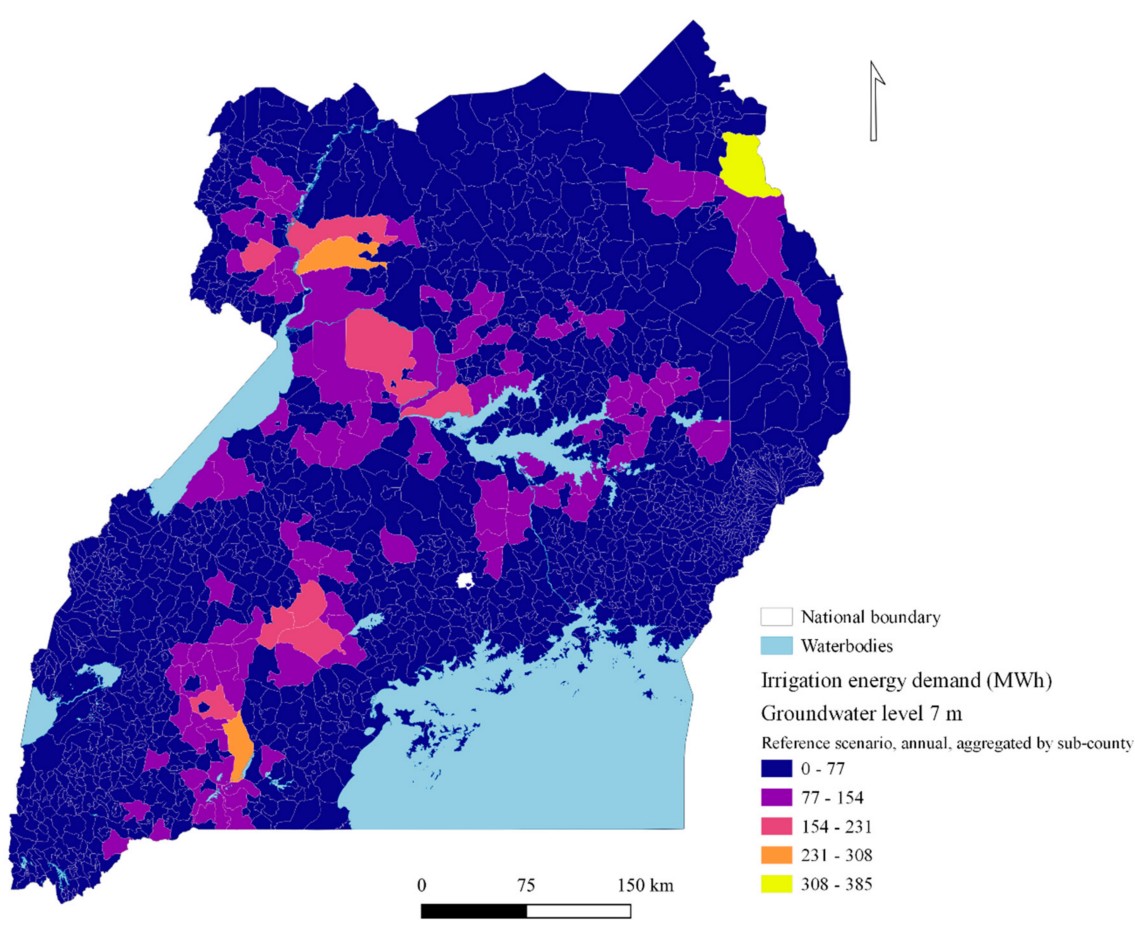

**Figure 9.** Aggregated annual energy demand (MWh) for irrigation water pumping per sub-county, reference scenario. The highest annual energy demand is linked to the area of the sub-county and the energy intensity per hectare.

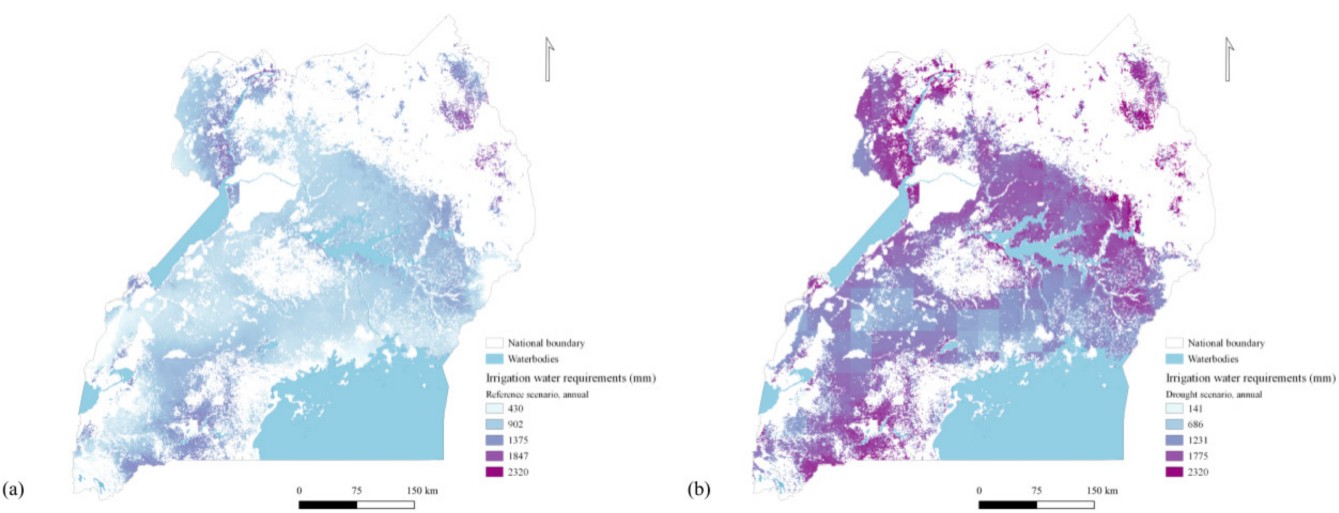

**Figure 10.** Estimated annual irrigation water requirements (mm) in the reference and drought scenario, respectively. The results are presented as follows: (**a**) annual irrigation water requirements in the reference scenario and (**b**) annual irrigation water requirements in the drought scenario.

Further, the relative increase in water demand is unevenly distributed across the year. As presented in Figure 11, a drought would require a substantially higher increase in irrigation water supply in April and May with an increase of 262% and 231%, respectively,

while the absolute highest increase in water demand is observed in September through November. By including such results in water availability assessments, the analysis could inform the planning of adaptive measures such as energy supply planning or the planning of cropping periods.

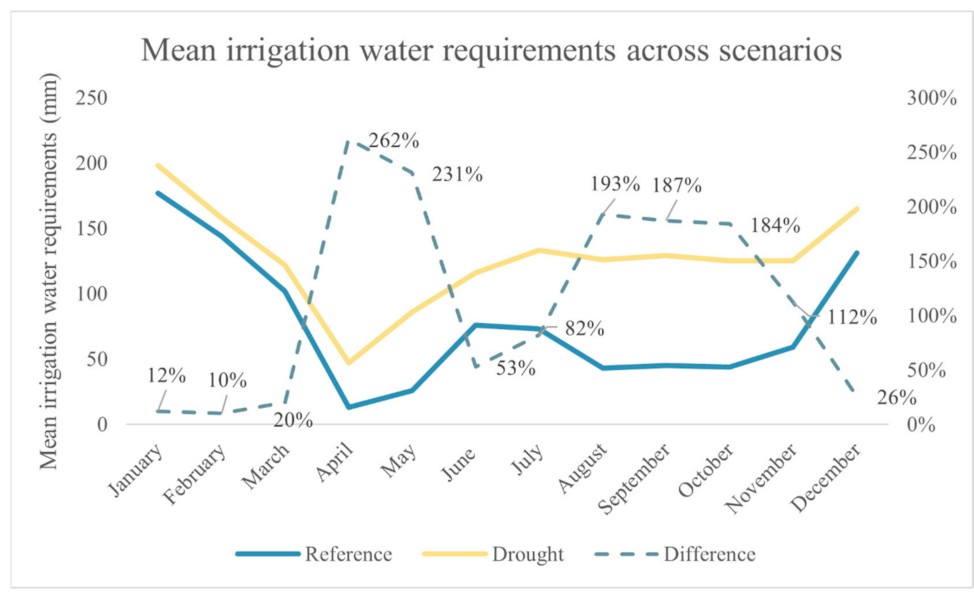

**Figure 11.** Monthly mean irrigation requirements (mm) in the reference and the drought scenario. The dashed line represents the difference in percentage between the reference and the drought scenario.

### 3.2.2. Power and Energy Demand, Drought Scenario

Similarly, the implications for the energy and power demand vary across the year as well as the study area. As presented in Figure 12, an increased power demand is observed throughout the year, yet with significantly higher absolute values.

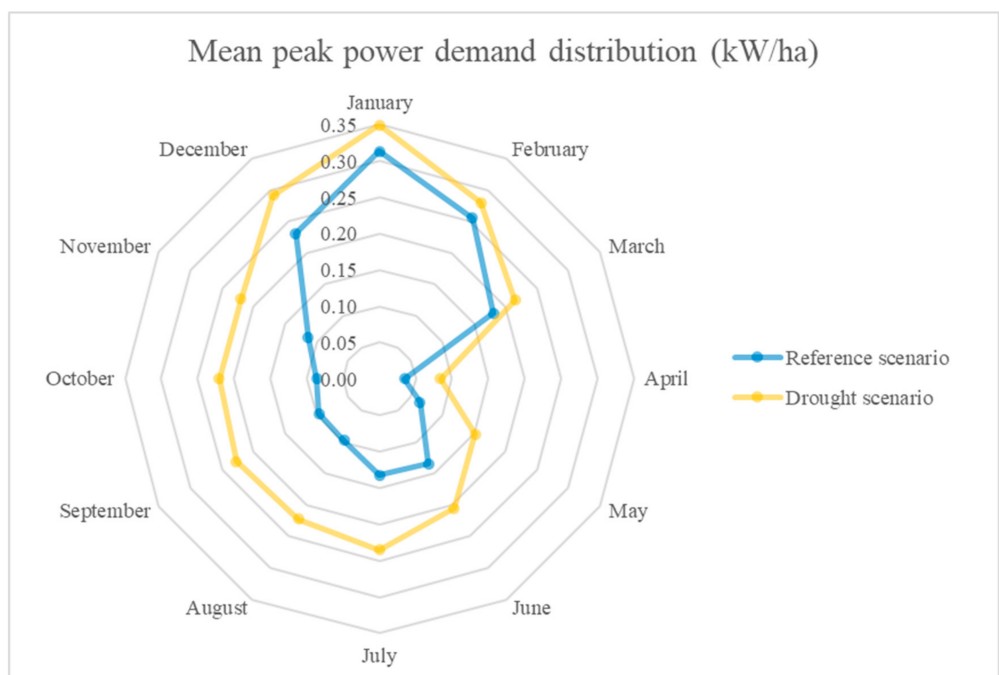

**Figure 12.** Mean peak power demand (kW/ha) distribution for irrigation water pumping across the year in the reference and drought scenario.

Figure 13 presents an overview of how the mean peak power demand throughout the year differs between the reference and drought scenario in the normally driest and wettest months of January and April, respectively. It demonstrates a particularly high increase in power demand in the normally wet months. As shown in Table 8, April experiences a 142% rise in demand. Even though there still is an increment in demand throughout the year, it is notably lower in the normally dry months: a 12% rise in January, for example. Overall, the average annual power demand rises from 0.14 to 0.23 kW/ha, corresponding to a 55% increase. An even higher increase is observed in the mean annual energy demand, growing by 67% from 0.18 to 0.30 MWh/ha between the reference and drought scenario. However, these numbers are measured on a country level and may vary significantly within specific regions.

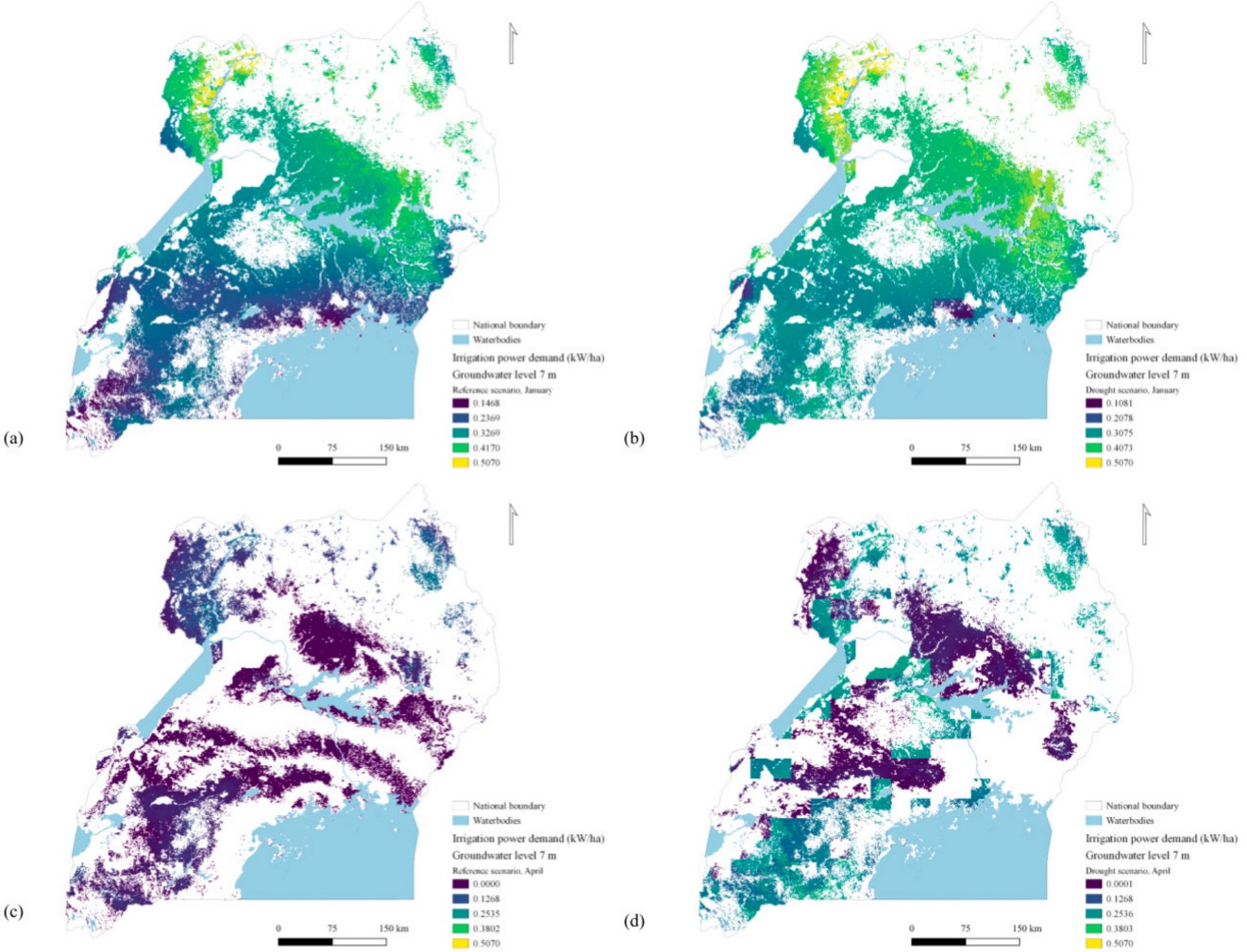

**Figure 13.** Peak power demand for irrigation water pumping (kWh/ha) compared between the reference and drought scenario in the normally wettest and driest months (January and April), respectively. The results are presented as follows: (**a**) January, reference scenario, (**b**) January, drought scenario, (**c**) April, reference scenario, and (**d**) April, drought scenario.

Comparing the power demand in April in the reference and drought scenarios (Figure 13c,d), a larger area in need for irrigation has emerged in the drought scenario, with demand appearances in the central region. Making the same comparison in January reveals a close to insignificant change in terms of an additional area in need for power for groundwater pumping—similarly as for other normally dry months.

As such, the scenario analysis results highlight that droughts could put significant increased pressure on the overall water supply. Also, the implications on the energy and power demand are likely to vary across the study area, where normally dry circumstances

are more severely impacted. In some months, the emergence of power for irrigation in areas not previously in need for irrigation could also be expected.

**Table 8.** The distribution of energy demand and peak power demand for irrigation water pumping across scenarios and time of the year.

| Scenario | Annual Energy Demand (MWh/ha) | | Mean Peak Power Demand (kW/ha) | | |
|---|---|---|---|---|---|
| | Max | Mean | April | January | Annual |
| Reference scenario | 0.77 | 0.18 | 0.03 | 0.31 | 0.14 |
| Drought scenario | 0.98 (+27%) | 0.30 (+67%) | 0.08 (+142%) | 0.35 (+12%) | 0.23 (+55%) |

## 4. Discussion

The aim of this study was to develop a methodology for the spatial estimation of the electricity demand for small-scale groundwater pumping, and to estimate the spatial irrigation water requirements, power, and energy demand for irrigation in Uganda.

The results provide indicative water, power, and energy demand estimates and their geographical distribution across Uganda under normal and abnormally dry circumstances. The results may be valuable to compare with other supply side models to better match energy supply and demand. The key messages from the analysis are summarized below.

### 4.1. Implications for Irrigation Requirements

Irrigation water requirements vary across the country as well as across the year. On a national level, the absolute irrigation water demand is estimated to be 90.4 thousand m$^3$. The highest need for irrigation is observed in the north and the south, where there is a demand for irrigation even in the normally driest months of April and May. The overall highest demand is seen in December through February, when the average demand reaches 445 mm.

Importantly, the estimates are based on historic climatic data, which, in the future, are likely to change patterns as an effect of climate change. For these reasons, the calculations are reiterated in a simulated year with abnormally low rainfall, based on the likelihood of droughts geographically across the study area. The results show that the peaks in the reference scenario tend to even out, as the normally wet months are more severely affected by droughts. Overall, the average irrigation water demand increases by 54% compared to the reference scenario, yet the geographical impacts vary. The southern and northern parts of the country are more severely affected and experience a higher relative rise in demand. This indicates that those regions may be in higher need of adaptive measures and need increased power supply in the future.

### 4.2. Implications for Energy Demand

As energy demand is directly linked to irrigation needs, this follows a similar pattern as irrigation water demand, geographically and throughout the year. The highest observed energy demand per hectare is observed in January in the northern region, which reaches 118 kWh. In the wetter months (April and May), energy demand does not surpass 48 kWh per hectare. In the drought scenario, there is a substantial increase in the total amount of irrigation water that needs to be delivered to the fields, whereby the average energy demand increases by 67%. The energy demand shows a high sensitivity to the groundwater level, increasing linearly with a factor of two, when the groundwater depth increases (see Supplementary Materials). Groundwater level analysis is, therefore, imperative for more detailed analysis of energy demand for irrigation and should be part of energy system planning and irrigation project planning. Further, the results also indicate an increased energy demand in the case of changing groundwater levels following drought events.

In terms of total annual energy demand, the aggregated energy demand varies significantly from one sub-county to another. The key reason for this is the varying acreages of the sub-counties, as the annual aggregated energy demand per sub-county typically increases

with its area, which can be observed in the results. The second influencing parameter is the estimated energy demand per hectare within the boundaries of the sub-county. The regional distribution of aggregated energy demand per year can help to identify clusters of high energy demand, which can further inform energy system planning.

### 4.3. Implications for Peak Power Demand

The most significant difference between the reference and drought scenario is observed in the normally wettest months. The reason for this is that in the driest months, there is already usually low precipitation, and the irrigation need is approaching $ET_C$. This means that a varying C has a lower impact on the results. Since the amount of rainfall initially is lower, this means that the absolute value of the decrease imposed by a decrease in C is lower compared to that in the case when the initial amount of rainfall is higher. By comparing the maximum observed peak power demands between the two scenarios, the difference is almost insignificant, which is different from the energy demand. This may indicate that the pump capacity is not necessarily a limiting factor to drought resilience; however, reliable energy supply is.

The differences between the power demand in April drought versus reference scenario is relatively large—peak power demand almost doubles. However, in some areas, a lack of demand is visible in the drought scenario, where there is a demand in the reference scenario. Simultaneously, a significant increase in demand is noted, particularly in the southern and most northern parts of the study area, but also scattered across other regions.

The sensitivity analysis to the time of operation implies high sensitivity to the power demand (see Supplementary Materials). This suggests that increased peak demand will stress the power supply system, meaning that the power supply will have to deliver a higher output. Such data could help inform the definition of system load curves for wider power models.

### 4.4. Key Limitations

The study is subject to a set of limitations that could be further improved in future research. A key aspect limiting the quality of the results is the accuracy and granularity of the input data. Considering its high impact on the results, the groundwater level data are of particular concern in this matter. More granular data on the groundwater table could significantly improve the accuracy of the results. Moreover, as this study focuses on groundwater pumping, potential areas where surface water could be used instead of groundwater are not accounted for. Future studies should, therefore, consider collecting spatial information on the proximity to surface water to further define areas where groundwater irrigation could be a plausible water supply solution.

Another limitation relates to the precipitation and associated variations in the crop coefficient. As the crop coefficient increases with the amount of precipitation, future studies could investigate the implications of this by using different crop coefficients across scenarios with varying precipitation.

Further, many of the input parameters can vary substantially, which can further affect the results in various ways. As the aim of this study is to generate indicative data to serve as an illustrative exercise, the impact of the potential range of variation of those parameters is not further investigated. However, this could be a relevant topic in future research efforts.

Although useful as an extreme case, the drought scenario assumes a lower-than-average precipitation across all months of the year. In many cases, this might only be the case during parts of the year. Nevertheless, the results from the drought scenario provide a useful example of extreme circumstances and could be particularly valuable to study on a monthly resolution.

The analysis can be enhanced in a number of ways. Analytical outputs might be improved with access to more up-to-date information and higher quality datasets. Future studies should further improve the introduced methodology by addressing those gaps, including the generation of more granular data on groundwater tables and the analysis

of the impact on varying input parameters. Conducting sensitivity analyses on specific input parameters allows for the identification of critical factors of which more accurate and granular data might be requested.

## 5. Conclusions

This study introduces a methodology for estimating the spatial distribution of irrigation water requirements for small-scale irrigation, through which power and energy demand estimates can be derived. As such, it contributes to current gaps in the existing literature by providing a methodology and data aimed at facilitating energy system planning.

Access to electricity can enable smallholder farmers to adopt more climate-smart agriculture and improve livelihoods through the application of small-scale irrigation. As such, small-scale irrigation can be instrumental in building resilience to climate change impacts, preserving biodiversity and ensuring food security. A precondition to a functional irrigation system is the reliable access to electricity. As the world is moving towards an increasingly electrified society and energy sector, energy planning requires accurate and granular data. In order to plan the power supply sector efficiently, the demand side needs to be properly understood. As the power sector is moving towards an increased penetration of variable renewable energy sources, the spatial distribution of supply and demand can facilitate the optimized planning of the expansion of the power sector.

While the analysis is a first of its kind in terms of scope, it provides the basis for an array of future studies, such as the involvement of multiple crops and the application of the methodological framework to other countries. Further, the study can be integrated to geospatial energy system models such as OnSSET and the Energy Access Explorer [64,65]. For instance, the results can be applied in energy system planning models to build more accurate system load curves, contributing to an improved understanding of the energy demand side. Further, outputs can be combined with geospatial data on variable renewable energy resources and power infrastructure to optimize the design of supply solutions. By applying the spatial dimension of the demand sector, critical regions can be identified and combined with additional data such as socio-economic parameters. By doing so, areas can be identified and prioritized in energy system planning. As such, this framework can contribute to better informed, evidence-based policy decisions related to the energy–water–agriculture nexus.

**Supplementary Materials:** The following are available online at https://www.mdpi.com/article/10.3390/ijgi10110780/s1, Figure S1: Key spatial datasets applied for the estimation of the irrigation water requirements. The datasets are presented in the following order: (a) Potential evapotranspiration (mm), (b) Precipitation in January (mm) (January is presented as an example, although data for all months of the year are applied in the calculations), (c) Soil clay content (%), (d) Soil water holding capacity (%), and (e) Soil sand content (%), Figure S2: Irrigation water requirements (mm) in the reference scenario, January through December, Figure S3: Irrigation water requirements (mm) in the drought scenario, January through December, Figure S4: Energy demand (kWh/ha) in the reference scenario, January through December, Figure S5: Energy demand (kWh/ha) in the drought scenario, January through December, Figure S6: Peak power demand (kW/ha) in the reference scenario, January through December, Figure S7: Peak power demand (kW/ha) in the drought scenario, January through December, Table S1: Annuel energy demand (MWh/ha) by groundwater level, Table S2: Peak power demand (kW/ha) in January and April, by time of operation ($t_{Op}$).

**Author Contributions:** Conceptualization, Anna Nilsson, Dimitrios Mentis and Alexandros Korkovelos; methodology, Anna Nilsson and Dimitrios Mentis; software, Anna Nilsson; validation, Anna Nilsson, Dimitrios Mentis, Alexandros Korkovelos and Joel Otwani; formal analysis, Anna Nilsson; investigation, Anna Nilsson; resources, Anna Nilsson, Dimitrios Mentis, Alexandros Korkovelos and Joel Otwani; data curation, Anna Nilsson; writing—original draft preparation, Anna Nilsson; writing—review and editing, Dimitrios Mentis, Alexandros Korkovelos and Joel Otwani; visualization, Anna Nilsson; supervision, Dimitrios Mentis and Alexandros Korkovelos; project administration, Anna Nilsson; funding acquisition, Dimitrios Mentis. All authors have read and agreed to the published version of the manuscript.

**Funding:** In kind contribution of office space by the Ministry of Energy and Mineral Development of Uganda.

**Institutional Review Board Statement:** Not applicable.

**Informed Consent Statement:** Not applicable.

**Data Availability Statement:** Not applicable.

**Acknowledgments:** This paper was co-supervised by Dimitrios Mentis with support from the World Resources Institute (WRI) and Alexandros Korkovelos with support from the Royal Institute of Technology (KTH). The author also wishes to thank the Ministry of Energy and Mineral Development of Uganda for the kind provision of office space.

**Conflicts of Interest:** The authors declare no conflict of interest.

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
