# Peer review of "A GIS-Based Approach to Estimate Electricity Requirements for Small-Scale Groundwater Irrigation"

_ijgi, doi:10.3390/ijgi10110780_

Round 1
Reviewer 1 Report
Dear authors, the article is well written and presented. I think it should be published in after fewer modifications. First, please try to reduce the size. Its way too lengthy. In order to reduce it size, please make some figures and data as supplimentry materials.
Good luck.
Author Response
Dear reviewer,
Many thanks for providing valuable feedback. Please find the responses below.
Feedback: First, please try to reduce the size. Its way too lengthy. In order to reduce it size, please make some figures and data as supplimentry materials.
The number of figures has been reduced. For the results, the authors have kept at least one figure per parameter in order to present the geographical varieties which is challenging to do with high accuracy in words. All figures (monthly and annual results) are added to the supplementary materials.
Reviewer 2 Report
The current manuscript discusses a GIS-based analysis for electricity use from irrigation need. The writing is generally good, and the reviewer has a few comments:
1) Parameters used in equations in this study are usually not fixed numbers. However, it appears that only fixed numbers are used. Please comment on the influence of the variation of parameters in the manuscript, or use a Monte Carlo approach instead.
2) The relation between electricity need and water demand is too simple. Not all irrigation water needs to be pumped. A lot of irrigation is done (if the water source is surface water) by gravity without pumping. Therefore, the author should consider elevation in this study. This is the reviewer’s main objection.
On the other hand, if pumping of groundwater is the focus, the author should isolate areas that solely rely on groundwater. The reviewer believes that most areas mix these two sources (surface and groundwater) for irrigation.
3) The definition of drought is not well-accepted. Please use a more common criterion for drought, such as the PDSI.
4) Line 478: NDCV?
5) Lines 561, 619, 631: Broken link to figures.
6) The manuscript does not specify the type of submission (Article, Review, Communication, etc.).
Author Response
Dear reviewer,
Many thanks for providing valuable feedback. Please find the author's responses below.
- Parameters used in equations in this study are usually not fixed numbers. However, it appears that only fixed numbers are used. Please comment on the influence of the variation of parameters in the manuscript, or use a Monte Carlo approach instead.
The authors address the sensitivity of input parameters on the analytical outputs in a sensitivity analysis found in the appendix as per the reviewer’s earlier comments. The authors indicate that the focus of this paper is the introduction of the methodological framework and not the specific number which should always be read with caution and with respect to the assumptions made.
- The relation between electricity need and water demand is too simple. Not all irrigation water needs to be pumped. A lot of irrigation is done (if the water source is surface water) by gravity without pumping. Therefore, the author should consider elevation in this study. This is the reviewer’s main objection. On the other hand, if pumping of groundwater is the focus, the author should isolate areas that solely rely on groundwater. The reviewer believes that most areas mix these two sources (surface and groundwater) for irrigation.
The authors thank the reviewer for this observation, but would like to highlight that this is beyond the scope of the study, which the authors mention across the article including the introduction, methodology and limitations and suggestions for future research. The study specifically investigates groundwater pumping, and suggests the analysis of surface irrigation for future studies.
- The definition of drought is not well-accepted. Please use a more common criterion for drought, such as the PDSI.
The main reason why the ADSI was used is because it reflects agricultural droughts. The PDSI is more general and captures better annual changes. As the study is looking particularly at agricultural water needs, and that at a monthly basis, the ADSI was considered more appropriate. This clarification has been added to section 2.5.2.
- Line 478: NDCV?
Thanks for identifying the typo – now corrected.
- Lines 561, 619, 631: Broken link to figures.
Thanks for the observation. Links have been recovered.
6) The manuscript does not specify the type of submission (Article, Review, Communication, etc.).
The type of submission (article) will be clarified when uploading the revised version.
Reviewer 3 Report
In general, it is an interesting paper. I have given several minor comments.
1、The abstract of the manuscript is too long and needs to be simplified. this is research paper, not a report, you should hit the main point and give the objective then.
2、Literature review should be extended, please review more relevant papers using this type of technique. and provide the citations properly.
3、The author should polish the language. Some syntax errors exist in the manuscript.
Author Response
Dear reviewer,
Many thanks for providing valuable feedback. Please find the author's responses below.
1、The abstract of the manuscript is too long and needs to be simplified. this is research paper, not a report, you should hit the main point and give the objective then.
The abstract has been shortened and simplified to the extent the authors find reasonable while still mentioning the key challenge and results.
2、Literature review should be extended, please review more relevant papers using this type of technique. and provide the citations properly.
We have added 3 new papers (see below). The literature review on GIS based assessments for ground water irrigation is rather limited. The authors have included the literature that is known to them.
- Falchetta, Giacomo et al. (2020) : M-LED: Multi-sectoral Latent Electricity Demand Assessment for Energy Access Planning, Working Paper, No. 009.2020, Fondazione Eni Enrico Mattei (FEEM), Milano
- Shirley, Y. Liu, J. Kakande, and M. Kagarura, “Identifying high-priority impact areas for electricity service to farmlands in Uganda through geospatial mapping,” J. Agric. Food Res., vol. 5, no. April, p. 100172, 2021, doi: 10.1016/j.jafr.2021.100172.
- Xie, H., Ringler, C., & Hossain Mondal, M. A. (2021). Solar or diesel: A comparison of costs for groundwater- fed irrigation in sub-Saharan Africa under two energy solutions. Earth's Future, 9, e2020EF001611. https://doi. org/10.1029/2020EF001611
3、The author should polish the language. Some syntax errors exist in the manuscript.
The authors have gone through the article and revised identified syntax errors.
Round 2
Reviewer 2 Report
Thank you for providing the response and a revised manuscript. Most questions have been solved, except for one. The reviewer understands that the authors wanted to limit the scope of this study on "groundwater demand" only. However, the extent of land that is irrigated by groundwater must be determined first. The reviewer might have missed such information in the manuscript. So far, the authors appear to assume that all fields use only groundwater. Such a questionable assumption will lead to unrealistic results.
Author Response
Dear reviewer,
Thank you for your feedback.
Under the key limitations, the authors are referring to this specific limitation and how this could be addressed by future research. More specifically “As this study focuses on groundwater pumping, potential areas where surface water could be used instead of groundwater are not accounted for. Future studies should therefore consider collecting spatial information on the proximity to surface water to further define areas where groundwater irrigation could be a plausible water supply solution”. Better defining the areas that could be covered by groundwater and/or surface water irrigation would require a better understanding of the surface water availability and the extent this could be made available. While this is an important component of a more holistic water supply assessment, it has been left out of scope from this study as defined already in the introductory sections.
This manuscript is a resubmission of an earlier submission. The following is a list of the peer review reports and author responses from that submission.
Round 1
Reviewer 1 Report
Dear Editor,
Thank you for giving me the opportunity to review this MS. There are no issues of conflicting interest, and I have no personal or professional affiliation with the authors.
This manuscript presents a study on A GIS Based Approach to Estimate Electricity Requirements for Small-scale Groundwater Irrigation.
This is a potentially good manuscript and needs major revision and re-evaluation. However, I appreciate to take these comments into account.
The following revisions are suggested:
- In abstract and introduction you mentioned an excessive details about SSA, however your study are outside SSA. So please focus on your study area and surroundings.
Introduction:
- The introduction is so long and has a lot of unnecessary details. Only one page for introduction is sufficient. Also, focus only on your study area and its surroundings, not SSA.
- Page 3, line 180; “…further maximised b through…” what do you mean by “b”?
Materials and methods:
- Page 6, line 245; “1.1.2. Farming system…” change the numbering to be “2.1.2. …”.
- Page 6, line 247; “Highland perennial farming systems in SSA ….” Delete this sentence and focus on your study area.
- Page 7, line 263 to 265; It is already known by any one that water is the most critical factor for agriculture, delete it.
- Page 7, line 276; this paragraph is not suitable with its subtitle (definition of the study area”. Make it first paragraph in “Materials and methods” or add it to the data used.
- Page 11, line 329; “…from the soil and the plant to the atmosphere.” Delete “the”.
Figures:
- For all figures add Coordinates, North direction, and scale, and write more details in cations. Also, try to add two or more maps in one figure as in figure 6, for example it is better to make figures from 7 to 11 together in one figure and number them as (a, b, …).
- In figure (6) number the maps into (a) and (b).
Notice: Attached file includes many corrections and suggestions that should be considered in further revision.

Author Response
Dear reviewer,
We are thankful for your valuable feedback. Please find our responses below in blue.
- In abstract and introduction you mentioned an excessive details about SSA, however your study are outside SSA. So please focus on your study area and surroundings.
We are thankful to reviewer 1 for his/her feedback. The introductory section has been reduced. However, Uganda is one of the countries in the SSA region (link) and as such the authors think it is critical to provide the broader context (especially if we want to think about the scalability of the introduced methodology).
Introduction:
- The introduction is so long and has a lot of unnecessary details. Only one page for introduction is sufficient. Also, focus only on your study area and its surroundings, not SSA.
Please see the response to the previous comment. - Page 3, line 180; “…further maximised b through…” what do you mean by “b”?
Thanks for noticing, the error has been corrected.
Materials and methods:
- Page 6, line 245; “1.1.2. Farming system…” change the numbering to be “2.1.2. …”.
Noted and changed. - Page 6, line 247; “Highland perennial farming systems in SSA ….” Delete this sentence and focus on your study area.
The author argues that this is relevant to the study considering that the study area is part of SSA. Please refer to the response to the first comment of the reviewer. - Page 7, line 263 to 265; It is already known by any one that water is the most critical factor for agriculture, delete it.
Deleted. - Page 7, line 276; this paragraph is not suitable with its subtitle (definition of the study area”. Make it first paragraph in “Materials and methods” or add it to the data used.
Figure 1 and the related paragraph has been moved as suggested. - Page 11, line 329; “…from the soil and the plant to the atmosphere.” Delete “the”.
Deleted.
Figures:
- For all figures add Coordinates, North direction, and scale, and write more details in cations. Also, try to add two or more maps in one figure as in figure 6, for example it is better to make figures from 7 to 11 together in one figure and number them as (a, b, …).
Figures have been updated and grouped as suggested. More information has been added to the captions. - In figure (6) number the maps into (a) and (b).
Figures has been numbered as suggested by the reviewer.
Notice: Attached file includes many corrections and suggestions that should be considered in further revision.
The author thanks the reviewer for providing additional suggestions. These have been addressed by the authors.
Reviewer 2 Report
I appreciate the Editor to give me a chance to review an interesting and valuable paper. I found some merits in the both methodology and results. In my opinion, this paper has a good potential to be published in the journal. However, I have also some concerns on the different parts of the manuscript. If the author(s) address carefully to the comments, I’ll recommend publication of the manuscript in the journal:
- Add some of the most important quantitative results to the Abstract.
- Add/Replace the name of the study area to the Keywords.
- Cite this recent useful paper as the reference of the first sentence of the Introduction to improve the literature and to show the importance of your work:
Energy Sustainability with a Focus on Environmental Perspectives
- Discuss the main reasons for the variations of the aggregated annual energy demand (GWh) per sub-county.
- How can extend the results in other regions with similar/different groundwater conditions?
- At the end of the manuscript, explain the implications and future works considering the outputs of the current study.
Author Response
Dear reviewer,
We thank you for your valuable feedback. Please find our responses below in blue.
- Add some of the most important quantitative results to the Abstract.
Key quantitative results have been added to the abstract. - Add/Replace the name of the study area to the Keywords.
Added - Cite this recent useful paper as the reference of the first sentence of the Introduction to improve the literature and to show the importance of your work:
Energy Sustainability with a Focus on Environmental Perspectives
Thanks, the paper has been cited.
- Discuss the main reasons for the variations of the aggregated annual energy demand (GWh) per sub-county.
A discussion around this has been added to the discussion (subsection 4.1.2) - How can extend the results in other regions with similar/different groundwater conditions?
The author has added an additional section on limitations (subsection 4.1.5) in the discussion where the relevance of accurate and granular groundwater data for future studies is discussed. A brief note on the expansion of the methodology to other study areas is also included in the conclusions. - At the end of the manuscript, explain the implications and future works considering the outputs of the current study.
Suggestions for future study areas have been added by the author in the section on limitations (subsection 4.1.5). The most relevant topics are further reiterated in the end of the conclusions section.
Reviewer 3 Report
The current manuscript discusses a GIS-based analysis for electricity use from irrigation needs. The writing is generally good, and the reviewer has a few comments:
- Since the topic is focused on electricity use, the reviewer suggests removing “water need” from line 16.
- Line 139: It is not common to capitalize “billion”.
- Line 276: Broken link. Please check for similar issues throughout the manuscript.
- Section 1.1.2: It would provide more information linking to the topic of this manuscript if the author can provide the water use trends for different agriculture systems.
- Sections 2.3.2-2.3.4: A lot of parameters defined in these sections can have a lot of variation. Using a fixed number can distort the result. The reviewer suggests defining a range for parameters, and apply an approach similar to the Monte Carlo approach to determine the range of possible outcomes.
- Section 2.4.1: Please provide figures of snapshots of the input spatial data.
- Equation 8: 1) Please define IWR (irrigation water requirement?), 2) The relation between electricity need and IWR is too simple. Not all irrigation water needs to be pumped. A lot of irrigation is done (if the water source is surface water) by gravity without pumping. Therefore, the author should consider elevation in this study. This is the reviewer’s main objection. On the other hand, if the pumping of groundwater is the focus, the author should isolate areas that solely rely on groundwater. The reviewer believes that most areas mix these two sources (surface and groundwater) for irrigation.
- Line 535: The definition of drought is not well-accepted. Please use a more common criterion for drought, such as the PDSI.
- Line 552: NDCV?
- Line 773: The reviewer does not understand why the PDSI is used here, instead of used for defining droughts?
- Figure 24: the caption is out of place.
Author Response
Dear reviewer,
We thank you for your valuable feedback. Please find our responses below in blue.
Comments and Suggestions for Authors
The current manuscript discusses a GIS-based analysis for electricity use from irrigation needs. The writing is generally good, and the reviewer has a few comments:
- Since the topic is focused on electricity use, the reviewer suggests removing “water need” from line 16.
We kindly as the reviewer to please specify. The author cannot find “water need” mentioned in the abstract.
- Line 139: It is not common to capitalize “billion”.
Edited. - Line 276: Broken link. Please check for similar issues throughout the manuscript.
The author kindly requests the reviewer to specify which link is referred to, i.e. reference, title? - Section 1.1.2: It would provide more information linking to the topic of this manuscript if the author can provide the water use trends for different agriculture systems.
A sentence has been added to explain that the majority of farmers in the SSA are dependent on precipitation for their agricultural activities. - Sections 2.3.2-2.3.4: A lot of parameters defined in these sections can have a lot of variation. Using a fixed number can distort the result. The reviewer suggests defining a range for parameters, and apply an approach similar to the Monte Carlo approach to determine the range of possible outcomes.
The authors agree with the reviewers concerns and acknowledges that the variation of these input parameters may affect the results to various extent. However, as the aim of the study study is to provide order-of-magnitude results for illustrative purposes, the authors suggest not to investigate this matter quantitatively in this study. Instead, the authors have highlighted this as a potential are of future research in the discussion section on key limitations (subsection 4.1.5). - Section 2.4.1: Please provide figures of snapshots of the input spatial data.
Noted. The author agrees that adding some of this data would be valuable. This would include the two following:
- Groundwater table
- Since the groundwater data suggest that the full study area belongs to the same category (0-7 m) i.e. there is no spatial differences across the study area, the author can therefore see no added value in presenting this figure.
- Irrigation water requirements
- This data is generated from a set of different parameters. Since the irrigation water requirements are presented in the results section, the author suggest to present parameters used to generate the irrigation water requirements. A figure presenting those have been added to section 2.3.
Equation 8: 1) Please define IWR (irrigation water requirement?), 2) The relation between electricity need and IWR is too simple. Not all irrigation water needs to be pumped. A lot of irrigation is done (if the water source is surface water) by gravity without pumping. Therefore, the author should consider elevation in this study. This is the reviewer’s main objection. On the other hand, if the pumping of groundwater is the focus, the author should isolate areas that solely rely on groundwater. The reviewer believes that most areas mix these two sources (surface and groundwater) for irrigation.
The authors thank the reviewer for this observation which is a known limitation of the study. As the study focuses on groundwater pumping, future studies should also consider spatial information on the proximity to surface water to further define areas where groundwater irrigation could be a plausible water supply solution. This has been highlighted in the discussion (subsection 4.1.5).
- Line 535: The definition of drought is not well-accepted. Please use a more common criterion for drought, such as the PDSI.
As explained in section 2.5.2, the drought is derived from the NDVI percentile index, developed by the African Flood and Drought Monitor. The index specifically targets agricultural drought and is based on the soil moisture content and the Normalized Differential Vegetation Index. The authors kindly asks the reviewer to please specify for which reasons this index is not accepted to define a drought. - Line 552: NDCV?
Thanks for noticing the error – changed to NDVI. - Line 773: The reviewer does not understand why the PDSI is used here, instead of used for defining droughts?
The main reason for this is that the PDSI is considered as a more suitable index to estimate the effects form climate change in the long term. This is considered more suitable for the socioeconomic analysis, while the drought scenario is more focused on direct implications for the agriculture. Nevertheless, the authors understand the reviewer’s concerns, and could change the methodology to stick to one drought definition should the reviewer insist. If so, please let the authors know. - Figure 24: the caption is out of place.
Caption has been revised.
Reviewer 4 Report
Dear authors,
The article is exceptionally long, seems like a dissertation. The way article is designed including figures is very confusing and misleading. Please make a precise study comprising of max. 25 pages for further evaluations. For my side need to be entirely re-written and submitted again.
Round 2
Reviewer 3 Report
Thanks for providing a revised manuscript. Unfortunately, one major problem with the original manuscript identified by the reviewer in the first round of review was not corrected in the revised manuscript. As the authors identified in the newly added 4.1.5, the current manuscript assumes that 100% of irrigation water is provided by groundwater, which is not nearly realistic. Not considering multiple sources of irrigation water is a major issue that must be addressed.